# Wind turbine wake dynamics subjected to atmospheric gravity waves: A measurement-driven large-eddy simulation study

Dachuan Feng<sup>1</sup> and Simon Watson<sup>1</sup>

<sup>1</sup>Faculty of Aerospace Engineering, Delft University of Technology, 2629 HS Delft, the Netherlands **Correspondence:** Dachuan Feng (D.Feng@tudelft.nl) and Simon Watson (S.J.Watson@tudelft.nl)

**Abstract.** Atmospheric gravity waves (AGWs) are large-scale wave-like flow structures commonly generated when atmospheric flows are vertically displaced by topography. These transient phenomena can significantly affect wind turbine outputs and loads; however, their influence on wake dynamics remains poorly understood, posing challenges for accurate wind farm modeling. In this study, we perform large-eddy simulation of wind turbines operating under an atmospheric condition recon-

- structed by assimilating lidar measurements of AGWs. Our results show that: (i) Low-frequency wake meandering becomes more pronounced owing to large-scale AGW flow structures and intensified smaller-scale turbulent structures. This enhanced meandering, combined with stronger turbulent mixing, accelerates mean wake recovery. (ii) Turbulence kinetic energy (TKE) spectrum in the wake region exhibits a peak Strouhal number of approximately 0.3, although the inflow spectrum peaks at significantly lower frequencies. This observation indicates that, under AGW conditions, wake turbulence generation follows a
- convective instability mechanism. Notably, faster wake recovery reduces wake shear, leading to lower amplification of TKE. Power analysis for three turbines arranged in a streamwise column further highlights the dominant role of convective instabilities. Large-amplitude, low-frequency power fluctuations observed at the most upstream turbine are significantly attenuated for downstream turbines, as low-frequency velocity fluctuations shift to higher frequencies in the far-wake regions. These findings add further insights into wake meandering and turbulence generation, offering guidance for modeling wind turbine and farm
- flows under non-stationary atmospheric conditions.

## 1 Introduction

Atmospheric gravity waves (AGWs) commonly occur when the atmosphere is vertically displaced by topographical features, such as mountains and coastlines (Nappo, 2012). These waves are trapped by the stable capping inversion layer aloft and propagate horizontally within the lowest 1-5 km of the troposphere (Durran, 2003). This transient phenomenon can cause fluctuations in wind speed experienced by wind farms, resulting in variations in power output and aerodynamic loading compared

tuations in wind speed experienced by wind farms, resulting in variations in power output and aerodynamic loading compared to conditions under mean synoptic forcing. In a wind farm, most turbines operate in the wake regions of upstream turbines. Therefore, understanding the response of turbine wakes to atmospheric phenomena, including AGWs, is critical for accurately modeling wind farm performance under realistic atmospheric conditions.

The influence of AGWs on wind farm performance has recently attracted considerable attention (Wilczak et al., 2019; 25 Xia et al., 2021; Draxl et al., 2021). Through spectral analysis of field measurements and mesoscale simulations, researchers

have observed that low-frequency oscillations in turbine and farm power production correlate with wind speed fluctuations associated with AGWs. More recently, Ollier and Watson (2022) conducted a parametric study using Reynolds-averaged Navier-Stokes simulations to investigate factors influencing AGW effects. Their findings highlight the importance of the wind farm's location within the wave cycle: wake losses in power production are mitigated during AGW peaks and exacerbated during AGW troughs.

Large-eddy simulation (LES) provides a robust method for generating detailed data on wind farm flows and outputs. To account for large-scale meteorological effects, mesoscale simulations (e.g., the Weather Research and Forecasting (WRF) model (Skamarock et al., 2008)) or field measurements (Allaerts et al., 2023; Quon, 2024) are often used to inform mesoscale forcing, which captures non-stationary atmospheric conditions such as diurnal thermal instability. Recently, Wise et al. (2024)

combined LES with the WRF model to study interactions between AGWs and wind farms. Their results revealed that the 35 passage of an AGW modulates the mesoscale environment, significantly impacting wind farm power production and structural loading.

Despite these advancements, several gaps remain. While prior studies have primarily focused on turbine outputs, the response of turbine wakes to AGWs remains poorly understood. Furthermore, it is unclear whether LES driven by field measurements

can accurately capture transient atmospheric phenomena like AGWs. In this work, we use measurement-driven LES to explore the potential effects of AGWs on wake dynamics. Our analysis focuses on two key wake phenomena: (1) low-frequency meandering motions, which govern wake expansion and recovery, and (2) turbulence generation due to wake shear, which enhances turbulent kinetic energy (TKE) within the wake region.

We introduce the American WAKE experimeNt (AWAKEN) field campaign which provides measurements of AGWs and the measurement-driven LES setup in Sect. 2. Then, we analyze the effects of AGWs on wake meandering, wake turbulence 45 generation, mean wake recovery, and power fluctuations in Sect. 3. Finally, we present our conclusions in Sect. 4.

# 2 Methods

#### 2.1 AWAKEN measurement

AWAKEN is a large-scale field campaign designed to obtain detailed observations of wind farm-atmosphere interactions, with the goal of advancing the understanding of wind farm physics and improving overall performance (Moriarty et al., 2020). 50 Figure 1 shows the schematic of the measurement sites and terrain features in the AWAKEN campaign. Given the prevailing southerly wind direction, site A1 serves as the inflow condition for the King Plains wind farm, which is the most instrumented wind farm. The west-east mountainous terrain in this region causes multiple AGW events, which have been identified from atmospheric measurements.

55 For this study, we focus on the AGW event on 8 June 2023, because its vertical extent spans the rotor layer of a wind turbine. Figure 2 shows the radar observations of this AGW event. The wind speed peaks and troughs elongated in the northwestsoutheast direction indicate large-scale wave-like structures flowing over multiple wind farms. These wave-like structures horizontally propagate from southwest to northeast with a wavelength of approximately 2.5-3 km.