# Peer review of "Wind turbine wake dynamics subjected to atmospheric gravity waves: A measurement-driven large-eddy simulation study"

_Wind Energy Science, 2025_

## Referee Comment (RC1)

**Summary**

This manuscript aims to dissect the effect of atmospheric gravity waves (AGWs) on wake dynamics, including both wake meandering and wake-generated turbulence. The author's use large-eddy simulation (SOWFA) to model a conventionally neutral boundary layer with the wind turbine(s) parameterized using an actuator disk with rotation. To account for AGWs, they use measurements from a lidar assimilated into the LES. The lidar profiles come from data obtained during the American Wake Experiment (AWAKEN). The main conclusions of the study are that AGWs increase wake meandering and wake recovery. In terms of power production, the leading turbine feels the brunt of the AGW with the wake then attenuating the energetic frequencies associated with the AGW. As a result, the effect of AGW fluctuations on power for downwind turbines is small. Overall, the paper is comprehensive for a single set of atmospheric conditions. Their methods do a great job of isolating AGW effects to arrive at their conclusions. I believe the manuscript could be improved with minor revisions including more detail and justification for modeling design choices.

**Specific Comments**

1. I do not believe that the AGWs shown in Fig. 2 are due to topography, but I also believe that the generation mechanism for the simulated AGWs is not relevant to the work. Unless the authors can provide compelling evidence that the AGWs are terrain-induced, any references stating that the AGWs are a result of topography should be removed or reworded. This starts with the first sentence of the abstract. The authors could state that there are a number of generation mechanisms for AGWs (which is why this is a relevant paper, because they can happen for so many different reasons). Lines 244-245 in the conclusion even briefly discuss how there are other trigger mechanisms.

In the second sentence of the abstract, the authors refer to AGWs due to topography as transient phenomena; however, mountain waves are typically stationary, which is mentioned in line 63. I do not believe that the second part of the sentence that begins on line 63 to be correct (unless a citation can be provided). Mountain waves are stationary and can be broken down by the Froude number (Stull, 2017). The characteristics of mountain waves depend on the mean flow and mountain characteristics and as a result are just stretched or compressed but do not advect.

All of this is just to say that for the manuscript, the generation mechanism is not important but that the author's should exercise caution in their justification/discussion of their simulated AGWs. **Reference:** Stull, R., 2017: "Practical Meteorology: An Algebra-based Survey of Atmospheric Science" -version 1.02b. Univ. of British Columbia: https://geo.libretexts.org/Bookshelves/Meteorology\_and\_Climate\_Science/Practica L\_Meteorology\_(Stull)/17%3A\_Regional\_Winds/17.7%3A\_Mountain\_Waves

- 2. The author's simulate a conventionally neutral boundary layer; however, the decision to use a CNBL needs to be justified. During the period in which the lidar data is extracted, I assume that some information about stability could be extracted from other instruments at AWAKEN. How AGWs affect wake characteristics could vary significantly in stable or unstable conditions and this should be discussed.
- 3. At the end of Section 3.1, the authors state that the presence of AGWs increases both horizontal and vertical meandering. Why they increase vertical meandering is relatively intuitive; however, the manuscript does not provide any discussion or analysis on *why* they increase horizontal meandering. Additionally, Figure 4 is a nice visualization, but it would be nice to be helpful to see the large-scale AGWs in the flow. The AGWs shown in Fig. 2 span for 10s of kilometers in the lateral dimension, but Fig. 4 only shows ~250m in the lateral dimension. It would be nice for a reader to qualitatively compare the simulated and observed flow.
- 4. In Fig. 8, the higher frequency peak at 0.1 corresponding to the Strouhal number of the atmosphere requires more discussion. Why is there a peak corresponding to the boundary layer thickness? Is this something that is observed or has been seen in other simulations that the authors can reference?
- 5. In Section 3.4, why are the turbines only separated by only 4D? This seems to be quite a small separation distance. In Fig. 2, the propagation direction of the AGWs is to the northeast and the separation distances look quite large for the wind farms for that wind direction. The only time 4D would be appropriate would be for a due east or due west AGW propagation direction. Ultimately, the findings in this subsection are quite insightful and I feel like it would be very helpful to see a similar setup but with further separation distances.

Also, considering that attenuation is discussed in this section, more plots of spectra are needed. It would be nice to see the TKE spectra for the inflow representing the downwind turbines or even for just the power signal. Then the attenuation can be quantified.

**Minor Comments**

- Line 21: I would suggest qualifying this statement with the caveat that it depends on the wind direction. Or at least clarifying that this is due to how wind turbines are sited within a wind farm.
- Line 53: The terrain at AWAKEN is not mountainous. In Fig. 1, the valleys and peaks are pronounced because of the colorbar. The greatest elevation difference looks like it is just over 100m over a distance of several kilometers.
- Figure 3: I appreciate that the contours are explained in the caption, but I think there needs to be a color bar because there are colors other than green and yellow and it is difficult to interpret values in between.
- Line 91: "onshore terrain" is too vague. State what the land is used for near AWAKEN. Is it agricultural? Or perhaps shrubland?
- Line 104: This claim should have a citation.
- Line 106: Is the turbine operating at 9 rpm specifically for this case? Typically, there is a range of rpm turbines operate at from cut-in to rated.
- Line 125: Please provide a citation for the definition of wake meandering for the interested reader.
- Figure 7: the legend entry for non-AGW almost makes it look like there is a third entry. Please rearrange the entries for clarity.
- Ine 184-185: delete "in the instantaneous wake flow".
- Figure 9: I would suggest using a different colorbar for velocity and TKE (Fig. 6).

---

## Author Comment (AC1)

We thank the reviewer for reviewing our manuscript and providing comments to improve our work. Below are the point-by-point comments, replies and changes.

**Specific Comments**

(1) I do not believe that the AGWs shown in Fig. 2 are due to topography, but I also believe that the generation mechanism for the simulated AGWs is not relevant to the work. Unless the authors can provide compelling evidence that the AGWs are terrain-induced, any references stating that the AGWs are a result of topography should be removed or reworded. This starts with the first sentence of the abstract. The authors could state that there are a number of generation mechanisms for AGWs (which is why this is a relevant paper, because they can happen for so many different reasons). Lines 244-245 in the conclusion even briefly discuss how there are other trigger mechanisms.

In the second sentence of the abstract, the authors refer to AGWs due to topography as transient phenomena; however, mountain waves are typically stationary, which is mentioned in line 63. I do not believe that the second part of the sentence that begins on line 63 to be correct (unless a citation can be provided). Mountain waves are stationary and can be broken down by the Froude number (Stull, 2017). The characteristics of mountain waves depend on the mean flow and mountain characteristics and as a result are just stretched or compressed but do not advect.

All of this is just to say that for the manuscript, the generation mechanism is not important but that the authors should exercise caution in their justification/discussion of their simulated AGWs.

Reference: Stull, R., 2017: "Practical Meteorology: An Algebra-based Survey of Atmospheric Science" -version 1.02b. Univ. of British Columbia:

https://geo.libretexts.org/Bookshelves/Meteorology\_and\_Climate\_Science/Practical\_Meteorology\_(Stull)/17%3A\_Regional\_Winds/17.7%3A\_Mountain\_Waves

**Reply**

We thank the reviewer for these constructive comments regarding the generation mechanisms and wave characteristics of the present AGW event. Regarding generation mechanisms, we agree with the reviewer that AGWs can be triggered by various atmospheric phenomena, including frontal systems, thunderstorms, and orographic effects. Regarding wave characteristics, we partially agree with the reviewer that mountain waves are stationary in theory. We however would like to note that, mountain waves can also advect, as observed in the work of Sato et al. [1]. Because the precise type of the present AGW event cannot be

confidently identified, we have removed the relevant statements in the revised manuscript to avoid over-speculation.

Although we do not ascertain the exact source of this AGW event, we emphasize that our atmospheric simulation, which is constructed by assimilating lidar measurements into LES model, captures the wind speed characteristics associated with AGWs. In Fig. 3 (renumbered as Fig. 2 in the revised draft), the time-height history of wind speed of AGWs from simulation shows overall agreement with measurement. To further quantify this comparison, we have added R-Fig. 1 (Fig. 3 in the revised draft) presenting the time series of wind speed at hub height. The results demonstrate that our simulation not only reproduces the large-scale oscillations observed in the lidar measurement data, but also resolves smaller-scale turbulence fluctuations. Such detailed turbulence information can provide a reliable inflow condition for turbine simulations.

R-Fig. 1: Time series of wind speed at hub-height from simulation for both cases, AGW and non-AGW, and measurement for the AGW case, AGW (measurement).

We would also like to emphasize that the present study is intended as a case study focusing on a specific AGW event. As the reviewer suggested, more accurate characterization of different AGW types is indeed essential for a comprehensive understanding of AGW-wind farm interactions. In future work, we plan to extend our study to other AGW types to further generalize our findings.

**Rivision**

- (i) We have added a description about AGW generation mechanisms in lines 17-18 in Introduction section.
- 'Atmospheric gravity waves (AGWs) commonly occur when the atmosphere is vertically displaced by topographical features, such as mountains and coastlines, or meteorological phenomena, such as fronts and thunderstorms (Stull, 1988; nappo, 2012).'
- (ii) We have added some text concerning different sources of AGWs and the necessity for future studies of these in lines 262-264 in Conclusions section.

'The present work is intended as a case-study focusing on a specific AGW event. Future study should incorporate AGW events originating from various sources and with different wavelengths to comprehensively understand their roles in turbine wake and wind farm flows.'

(iii) We have added R-Fig. 1 as Fig. 3 and added lines 100-102 to clarify imulation-measurement agreement.

'To further quantify these comparisons, we show in Fig.3 wind speed time series at the hub-height. The results indicate that our simulation not only captures the large-scale wavy oscillations observed in the measurements, but also resolves smaller-scale turbulent fluctuations.'

(2) The authors simulate a conventionally neutral boundary layer; however, the decision to use a CNBL needs to be justified. During the period in which the lidar data is extracted, I assume that some information about stability could be extracted from other instruments at AWAKEN. How AGWs affect wake characteristics could vary significantly in stable or unstable conditions and this should be discussed.

**Reply**

We agree with the reviewer that thermal stability can significantly affect wake characteristics. In our simulation, we assimilated the AGW wind speed profile from the AWAKEN measurements, while the temperature profile was simplified as a CNBL. This simplification was necessary because temperature measurement data were not available for the present AGW event. Consequently, the atmospheric condition in our simulation is neutral, whereas in reality it could have been either neutral or thermally stable. We acknowledge this limitation and will address the simulation of AGWs under non-neutral thermal stratification in our future work.

Despite the absence of temperature measurements, our approach still captures the turbulence characteristics of AGWs and thus provides a reliable inflow condition for turbine simulations, as also discussed in our response to Specific Comment (1). Moreover, the observed increase in turbulence levels under AGW conditions is consistent with the findings of Wise et al. [2], who simulated another AGW event from AWAKEN using a WRF-LES model with both velocity and temperature data.

**Rivision**

We have added clarification on stability effects in lines 80-82.

'Such a simplification of thermal stability condition is necessary due to lack of temperature measurement data. Consequently, the atmospheric condition in our simulation is neutral, whereas in reality it could have been either neutral or

(3) At the end of Section 3.1, the authors state that the presence of AGWs increases both horizontal and vertical meandering. Why they increase vertical meandering is relatively intuitive; however, the manuscript does not provide any discussion or analysis on why they increase horizontal meandering.

Additionally, Figure 4 is a nice visualization, but it would be nice to be helpful to see the large-scale AGWs in the flow. The AGWs shown in Fig. 2 span for 10s of kilometers in the lateral dimension, but Fig. 4 only shows ~250m in the lateral dimension. It would be nice for a reader to qualitatively compare the simulated and observed flow.

**Reply:**

Similarly to Fig. 8, R-Fig. 2 shows the wake (solid lines) and inflow (dotted lines) spectra for three components of velocity fluctuations (left: streamwise u', middle: spanwise v', right: vertical w').

R-Fig. 2: Wake (solid lines) and inflow (dotted lines) spectra for streamwise fluctuations u' (left), spanwise fluctuations v', (middle), vertical fluctuations w' (right).

For St

R-Fig. 3: Large-scale streamwise-elongated structures in wall turbulence [6].

In the present study, large-scale motions in the atmospheric flow are bounded by the inversion layer, which is indicated by the yellow arrow in Measurement (AGW) in Fig. 3 (renumbered as Fig. 2 in the revised draft). Accordingly, a low-frequency peak at  $St \approx 0.1$  is observed, corresponding to an inversion layer height of approximately 1.25 km.

**Revison**

We have added an explanation concerning the frequency peak corresponding to the inversion layer in lines 184-189.

'For wall-bounded flows at high Reynolds numbers (e.g. atmospheric boundary layers), Smits et al. (2011) reported that the pre-multiplied TKE spectra in the logarithmic and outer regions exhibit a large-wavelength/low-frequency peak associated with the boundary-layer thickness (i.e., the inversion-layer height in the atmospheric boundary layer). This peak arises from the presence of large-scale, streamwise-elongated structures with characteristic length scales in the order of the boundary-layer thickness (Hutchins et al., 2012).'

(5) In Section 3.4, why are the turbines only separated by only 4D? This seems to be quite a small separation distance. In Fig. 2, the propagation direction of the AGWs is to the northeast and the separation distances look quite large for the wind farms for that wind direction. The only time 4D would be appropriate would be for a due east or due west AGW propagation direction. Ultimately, the findings in this subsection are quite insightful and I feel like it would be very helpful to see a similar setup but with further separation distances.

Also, considering that attenuation is discussed in this section, more plots of spectra are needed. It would be nice to see the TKE spectra for the inflow representing downwind turbines or even for just the power signal. Then the attenuation can be quantified.

**Reply**

The reviewer is correct that the turbine spacing of 4D in our three-turbine simulation is smaller than the spacings typically used in the AWAKEN wind farms. We intentionally adopted 4D spacing to ensure that the downstream turbines remain within the wake region of the upstream turbines, so that their power fluctuations can serve as indicators of upstream wake characteristics.

As suggested by the reviewer, we have performed an additional three-turbine simulation with 8D spacing. We also add spectra of power in Fig. 13, together with the power time series. For clarity, here we show (i) TKE spectra from the single-turbine simulation in R-Fig. 4 and (ii) time series and spectra of turbine power from the three-turbine simulations in R-Fig. 5.

R-Fig. 4 shows TKE spectra for inflow region (dotted lines) and downstream 4D (left) and 8D (right) wake region (solid lines) from the single-turbine simulation. In both AGW and non-AGW cases, the wake spectra exhibit a dominant peak at St  $\approx 0.3$ . Such a frequency peak arises from a convective shear-instability mechanism that dominates far-wake dynamics, which generates turbulent kinetic energy at 0.1

R-Fig. 4: Wake (solid lines) and inflow (dashed lines) spectra at downstream 4D (left) and 8D (right) from the single-turbine simulation.

R-Fig. 5 shows the time series (left) and spectra (right) of turbine power for the three-turbine simulations with 4D (top) and 8D (bottom) spacings. For 4D spacing, the presence of AGWs induces large-scale power oscillations at the first turbine (T1), which are strongly attenuated at the downstream turbines (T2 and T3). For 8D spacing, the attenuation of power oscillations is weaker, and T2 still exhibits visible peaks with a time delay relative to T1. The difference in power attenuation between 4D and 8D spacing is also evident in the corresponding spectra. This behavior is because, as we discussed in R-Fig. 4, the shear instability mechanism that damps low-frequency velocity fluctuations becomes weaker at further downstream.

R-Fig. 5: Time series (left) and spectra (right) of turbine power for the three-turbine simulations with 4D (top) and 8D (bottom) spacings during the AGW event.

From these comparisons, we conclude that a 4D turbine spacing is appropriate for ensuring that downstream turbines are located within the wake region of upstream turbines, thereby highlighting wake-turbine interactions.

**Revison**

- (i) We have added 8D wake spectra for the single-turbine simulation in Fig. 8 (as also shown in R-Fig. 4) and discussed the downstream evolution of wake spectra in lines 194-195.
- 'For both the AGW and non-AGW cases, such a frequency peak becomes less prominent in 8D downstream, because wake recovery has largely weakened shear instabilities at this region.'
- (ii) We have added the power curves for the three-turbine simulation with 8D spacing in Fig. 11 (as also shown in R-Fig. 5) and discussed the power attenuation in lines 230-235.

'Figure 11 shows time series (left) and spectra (right) of turbine power for the three-turbine simulations with 4D (top) and 8D (bottom) spacings. For 4D spacing, the presence of AGWs induces large-scale power oscillations at the first turbine (T1), which are strongly attenuated at the downstream turbines (T2 and T3). For 8D spacing, the attenuation of power oscillations is weaker, and T2 still exhibits visible peaks with a time delay relative to T1. The difference in power attenuation between 4D and 8D spacing is also evident in the corresponding spectra. This behavior is because, as we showed in Fig. 8, the shear instability mechanism that damps low-frequency velocity fluctuations becomes weaker at further downstream.'

**Minor Comments**

(1) Line 21: I would suggest qualifying this statement with the caveat that it depends on the wind direction. Or at least clarifying that this is due to how wind turbines are sited within a wind farm.

**Reply**

We agree with the reviewer that whether turbines in a wind farm are waked depends on wind direction.

**Revison**

We have rewritten lines 21-22 to clarfy this point.

'In a wind farm, wind turbines can fully or partially operate in the wake regions of those upstream, depending on the wind direction.'

(2) Line 53: The terrain at AWAKEN is not mountainous. In Fig. 1, the valleys and peaks are pronounced because of the colorbar. The greatest elevation difference looks like it is just over 100m over a distance of several kilometers.

**Reply**

According to Debnath et al. [7], the terrain is a fluviatile plain with a gradual west-to-east slope.

**Revison**

In the revised draft, we have removed 'The west-east mountainous terrain in this region causes multiple AGW events' and added 'The terrain is fluviatile plain with a gradual west-to-east slope (Debnath et al., 2022).' in the caption of Fig. 1.

(3) Figure 3: I appreciate that the contours are explained in the caption, but I think there needs to be a color bar because there are colors other than green and yellow and it is difficult to interpret values in between.

**Revison**

We have replotted Fig. 3 (shown here as R-Fig. 6 and renumbered as Fig. 2 in the revised draft) and added a colorbar for clarity.

R-Fig. 6: Flow chart of the present measurement-driven LES study.

(4) Line 91: "onshore terrain" is too vague. State what the land is used for near AWAKEN. Is it agricultural? Or perhaps shrubland?

**Reply**

As have been noted in our reply to Minor Comment (2), the terrain at AWAKEN is a fluviatile plain with a gradual west-to-east slope.

**Revison**

We have removed 'onshore terrain' and added 'The terrain is fluviatile plain with a gradual west-to-east slope (Debnath et al., 2022).' in the caption of Fig. 1.

(5) Line 104: This claim should have a citation.

**Reply**

We have added references in the revised draft to show that, beyond three rotor diameters downstream, the present actuator-disk model is consistent with both wind-tunnel experiments [8] and actuator-line simulations [9].

**Revison**

We have added the reference from Wu & Porté-Agel [8] and Stevens et al. [9] in lines 110-112.

'While the effects of the nacelle and tower are neglected, this method has demonstrated good agreement with wind tunnel measurements and high-fidelity numerical simulations in the far wake region (Wu and Porté-Agel, 2011; Stevens et al. 2018), which primarily influences wind farm flow characteristics.'

(6) Line 106: Is the turbine operating at 9 rpm specifically for this case? Typically, there is a range of rpm turbines operate at from cut-in to rated.

**Reply**

Yes, we simplified the turbine operating condition to a constant 9 rpm, as our study is a case study of single-turbine wake dynamics rather than a simulation of the exact AWAKEN wind farm.

**Revison**

We have discussed the simplification of turbine operating condition in lines 112-115.

'The turbine operates at a constant rotational speed of nine rotations per minute (9 rpm). This simplified operating condition is used because the present work serves as a preliminary investigation of a single turbine, rather than a detailed simulation of the exact AWAKEN wind farm.'

(7) Line 125: Please provide a citation for the definition of wake meandering for the interested reader.

**Revison**

We have added the work of Ainslie [10] as a reference in lines 134-135.

'Wake meandering refers to large-scale oscillations of the wake flow driven by low-frequency spanwise and vertical velocity fluctuations in the atmospheric flow (Ainslie, 1988).'

(8) Figure 7: the legend entry for non-AGW almost makes it look like there is a

third entry. Please rearrange the entries for clarity.

**Revison**

We have modified the non-AGW entries as suggested by the reviewer.

(9) Line 184-185: delete "in the instantaneous wake flow".

**Revison**

We have deleted these words.

(10) Figure 9: I would suggest using a different colorbar for velocity and TKE (Fig. 6).

**Revison**

We agree with the reviewer, and we have used different colormaps for velocity (see Fig. 9) and TKE (see Fig. 6) in the revised draft.

**References**

- [1] Sato, K., Tateno, S., Watanabe S., and Kawatani Y., 2012. Gravity wave characteristics in the Southern Hemisphere revealed by a high-resolution middle-atmosphere general circulation model, J. Atmos. Sci., 69, pp.1378-1396.
- [2] Wise, A.S., Arthur, R.S., Abraham, A., Wharton, S., Krishnamurthy, R., Newsom, R., Hirth, B., Schroeder, J., Moriarty, P. and Chow, F.K., 2025. Large-eddy simulation of an atmospheric bore and associated gravity wave effects on wind farm performance in the southern Great Plains, Wind Energy Science, 10(6), pp.1007-1032.
- [3] Larsen, G. C., Madsen, H. A., Thomsen, K., and Larsen, T. J., 2008. Wake meandering: a pragmatic approach, Wind Energy: An International Journal for Progress and Applications in Wind Power Conversion Technology, 11, pp.377–395.
- [4] Feng, D., Li, L. K., Gupta, V., and Wan, M., 2022. Componentwise influence of upstream turbulence on the far-wake dynamics of wind turbines, Renewable Energy, 200, pp.1081–1091.
- [5] Smits, A. J., McKeon, B. J. and Marusic, I., 2011. High–Reynolds number wall turbulence, Annual Review of Fluid Mechanics, 43(1), pp.353-375.
- [6] Hutchins, N., Chauhan, K., Marusic, I., Monty, J. and Klewicki, J., 2012. Towards reconciling the large-scale structure of turbulent boundary layers in the atmosphere and laboratory, Boundary-layer meteorology, 145(2), pp.273-306.

- [7] Debnath, M., Scholbrock, A. K., Zalkind, D., Moriarty, P., Simley, E., Hamilton, N., Ivanov, C., Arthur, R. S., Barthelmie, R., Bodini, N., et al., 2022. Design of the American Wake Experiment (AWAKEN) field campaign, Journal of Physics: Conference Series, vol. 2265, p.022058.
- [8] Wu, Y.-T. and Porté-Agel, F., 2011. Large-eddy simulation of wind-turbine wakes: evaluation of turbine parametrisations, Boundary-layer meteorology, 138, pp.345 366.
- [9] Stevens, R.J., Martínez-Tossas, L.A. and Meneveau, C., 2018. Comparison of wind farm large eddy simulations using actuator disk and actuator line models with wind tunnel experiments. Renewable energy, 116, pp.470-478.
- [10] Ainslie J. F., 1988. Calculating the flowfield in the wake of wind turbines, Journal of Wind Engineering Industrial Aerodynamics, 27, pp.213–224.

---

## Author Comment (AC2)

We thank the reviewer for reviewing our manuscript and providing comments to improve our work. Below are the point-by-point comments, replies and changes.

(1) Extent to which the chosen interval represents flow at the AWAKEN site and more generally. Source of data should be given (line 54). The paper identifies use of location A1 and of 08 June 2023 only with the justification that the vertical extent of the gravity wave spans the rotor layer of the wind farm. It would help the reader to show how often these types of condition occur at the AWAKEN site - if only very rarely then is this relevant for design?, if regularly then what range of AGW vertical extents occur? were similar observations (phase lagged) obtained at down-wind sites (e.g. H on Figure 1 map) or does vertical extent differ over the streamwise spacing between turbine rows? More generally atmospheric gravity waves can occur at other sites and locations so some discussion on how the conditions represent AGW conditions at wind farm sites more generally (e.g. in terms of ABL thickness and AGW wavelength - mentioned to be 2 km in LES (line 176), compared to 2.5-3 km measured (line 58)? - and amplitude, not only site roughness which is not the only important factor between sites).

**Reply**

We thank the reviewer for raising questions that are crucial for clarifying both the motivation and the limitations of our work.

**(i) Source of measurement data**

The AGW event analyzed in this study was first identified from horizontal scans obtained by X-band radars (shown in Fig. 2, rearranged as the right panel of Fig. 1 in the revised manuscript). The detailed wind field of AGWs was then measured using a scanning Doppler lidar located at site A1. The top-left panel of Fig. 3 (renumbered as Fig. 2 in the revised manuscript) shows the time-height history of wind speed with a temporal resolution of ~6 s and vertical spacing of ~10 m. These high-resolution measurements are assimilated to capture the transient features of AGWs.

**(ii) AGW observations at the AWAKEN site**

In June 2023, four AGW events with similar wave periods of approximately 600 s were observed. Their vertical extent ranged from the surface up to ~3 km above ground level. R-Fig. 1 shows the time-height evolution of the three velocity components for the June 8 event, which is the focus of our study. In the vertical velocity component, similar large-scale wavy oscillations are observed at site A (upstream of the wind farm) and site H (downstream). Such transient atmospheric phenomena represent non-idealized atmospheric conditions that should be considered in wind farm design and operation. Accurate modeling of these phenomena is therefore important for real-time wind farm simulations.

R-Fig. 1: Three velocity components of AGWs at site A (left) and H (right).

**(iii) AGW wavelength**

We confirm that the AGW wavelength reported in line 176 (simulation) is consistent with that mentioned in line 58 (measurement). As shown in Fig. 8, the spectral peak at St  $\approx 0.05$  corresponds to a characteristic length scale of  $\sim 20$  turbine diameters ( $\sim 2520$  m), which agrees well with the wavelength observed from the radar measurements in Fig. 2.

**(iv) Limitations of our work**

We acknowledge the reviewer's point that AGWs vary in wavelength and in relation to atmospheric boundary layer (ABL) depth. This variability arises because AGWs can be triggered by multiple atmospheric processes, including frontal systems, thunderstorms, and orographic effects. We also agree that the vertical extent of AGWs may be modified by the blockage effect of wind turbine arrays, depending on the turbine spacing. Nevertheless, our study is intended as a preliminary investigation into how an observed AGW event influences single-turbine wake dynamics. Future work will extend this analysis to a wider range of atmospheric conditions and turbine layouts.

**Revison**

- (i) We have corrected all descriptions on the AGW wavelengths as 'approximately 2.5 km'.
- (ii) We have added the source of AGW data in lines 54-56.

'Multiple AGW events have been identified from horizontal scanning by X-band radars and vertical profiling by scanning Doppler lidars. The high-resolution lidar measurements are used in our data assimilation for capturing transient features of AGWs.'

(iii) We have added some text concerning different sources of AGWs and the necessity for future studies of these in lines 262-264 in Conclusions section.

'The present work is intended as a case-study focusing on a specific AGW event. Future study should incorporate AGW events originating from various sources and with different wavelengths to comprehensively understand their roles in

**turbine wake and wind farm flows.'**

(2) Extent to which the indirect profile assimilation method reproduces the LIDAR measurements of the selected AGW event. The three frames shown on left hand side of Figure 3 compare LIDAR measurements of time varying onset velocity to the simulated conditions. Lines 94-96 comment that "the present LES not only captures the low-frequency wind speed oscillations by the AGW event but also resolves turbulence structures with higher spatio-temporal resolution'. Whilst the simulation seems to capture the period of the selected AGW event there seem to be other differences that are not mentioned; for example the LES shows larger maximum velocity, possibly larger minimum velocity, change of turbulence over 0.5

R-Fig. 2: Time series of wind speed at hub-height from simulation for both cases, AGW and non-AGW, and measurement for the AGW case, AGW (measurement).

Regarding mesh resolution and SGS model, our selections follow previous studies of idealized atmospheric boundary layers [1,2], i.e., the non-AGW case in our work. Fig. 8 in the original draft shows that the main difference between the AGW and non-AGW cases (dotted lines) is that the AGW case exhibits higher turbulent kinetic energy at relatively low frequencies, St

R-Fig. 3: Flow chart of the present measurement-driven LES study.

Regarding the different TKE levels, the higher TKE observed in the AGW case is expected, as the large-scale wavy oscillations contribute additional energy at relatively large characteristic length scales. The goal of our work is to examine how such transient atmospheric inflow conditions differ from the idealized atmospheric boundary layer (non-AGW case) in their influence on wake dynamics.

**Revison**

We have added the inflow profile for the non-AGW case in Fig. 2 (herein R-Fig. 3) and its corresponding description in lines 98-100.

'This vertical wind profile differs significantly from that for the non-AGW case, where wind speed typically increases monotonically with height above the ground.'

(4) Choice of turbine modelled. Please summarise the differences and similarities between the deployed GE 2.8 MW turbine and the NREL 5 MW reference turbine to explain why this substitution was made and highlight the implications of any differences of diameter, hub-height and operating characteristics.

**Reply**

Regarding the geometric features, the differences between the GE 2.8 MW turbine and the NREL 5 MW reference turbine are minor: rotor diameter of 127 m vs. 126 m, and hub height of 88.5 m vs. 90 m. For the operating condition, we simplified the rotational speed to a constant 9 rpm.

Because detailed design data of the GE 2.8 MW turbine are not publicly available, we used the NREL 5 MW reference turbine as a substitute. As our study focuses

on single-turbine wake dynamics rather than replicating the exact AWAKEN wind farm, we consider this substitution appropriate for the scope of our work.

**Revison**

We have added above discussions in lines 105-108.

'This open-source turbine model is used as a proxy for the 2.8-MW General Electric turbines deployed at the King Plains wind farm. Regarding the geometric features, the differences between the GE 2.8 MW turbine and the NREL 5 MW reference turbine are minor: rotor diameter of 127 m vs. 126 m, and hub height of 88.5 m vs. 90 m, respectively.'

(5) Turbine modelling approach. Reference needed for statement on line 104-105 re model choice previously demonstrating good agreement. Since the focus of this study is on locations up to 8D downstream (and for Figures 8 and 12 at 4D downstream) please clarify that the previous demonstration of good agreement for far-wake predictions relates to comparable distances.

**Reply**

We have added references in the revised manuscript to show that, beyond three rotor diameters downstream, the present actuator-disk model is consistent with both wind-tunnel experiments [3] and actuator-line simulations [4].

**Revison**

We have added the work of Wu & Porté-Agel [3] and Stevens et al. [4] as references in lines 110-112.

'While the effects of the nacelle and tower are neglected, this method has demonstrated good agreement with wind tunnel measurements and high-fidelity numerical simulations in the far wake region (Wu and Porté-Agel, 2011; Stevens et al. 2018), which primarily influences wind farm flow characteristics.'

(6) Meandering results. The analysis focuses on the streamwise increase of amplitude of meandering of the wake center. To relate the observed variations to the two mechanisms identified (lines 143-145) it would be helpful to show that this meandering of wake center is occurring at the AGW period (~10 mins as Fig 3?), and to show the turbulence length-scales (which are not currently stated in the manuscript), or corresponding time-scales, for each case.

**Reply**

We plot R-Fig. 4 to show the time history of the spanwise (left) and vertical (right) wake center for both AGW and non-AGW cases during the AGW event. The wake center locations are obtained using a two-dimensional Gaussian fit to the instantaneous wake profile at six rotor diameters downstream. Gaps appear at

some time steps, particularly in the AGW case, because the wake is too turbulent to be reliably fitted. In both directions, the magnitudes of wake-center deflections are clearly larger in the AGW case. In the spanwise direction (left panel), the AGW case exhibits distinct large-scale oscillations.

R-Fig. 4: Time history of spanwise (left) and vertical (right) wake centers for AGW and non-AGW cases.

Also, we note that the spectra shown in Fig. 8 can indicate the turbulence length scales: the inverse of the Strouhal number corresponds to the wavelength normalized by the rotor diameter.

**Revison**

(i) We have added clarification on large-scale wake center deflections in lines 143-147.

'The wake centers are determined by first filtering the instantaneous wake-deficit flow field with a spatial filter spanning three rotor diameters to isolate meandering motions. The filtered wake deficit is then fitted to a two-dimensional Gaussian profile at each downstream location, following the method described by Trujillo et al. (2011). The location of the maximum wake deficit is taken as the wake center. In both directions, the magnitudes of wake-center deflections are found to be larger for the AGW case, as evident in Fig. 4.'

(ii) We have added how Strouhal number indicates turbulence length scales in lines 177-178.

'Note that the inverse of Strouhal number corresponds to wavelength normalized by the rotor diameter, indicating the characteristic turbulence length scales.'

(7) TKE spectra analysis. This is interesting, particularly the peak sustained at Strouhal Number  $\sim 0.3$ . However, the lack of peak at St  $\sim 0.05$  in the wake in AGW case seems to indicate that there are not variations in the wake at the AGW period; does this affect line 141-142? Could the same type of spectra be shown for 8D also to better understand whether the same spectral content persists as the amplitude of meandering increases into the far wake? Is there any explanation available for the higher harmonics observed in AGW case?

**Reply**

We thank the reviewer for raising this important point. Lines 141–142 should be corrected to state that the enhancement of wake meandering is primarily caused by the increase in inflow turbulent kinetic energy. The explanation is as follows.

We show in R-Fig. 5 the spectra of wake (solid lines) and inflow (dashed lines) at 4D (left) and 8D (right) downstream from the single-turbine simulation. In both AGW and non-AGW cases, the wake spectra exhibit a dominant peak at St  $\approx 0.3$ . Such a frequency peak arises from a convective shear-instability mechanism that dominates far-wake dynamics, which generates turbulent kinetic energy at 0.1

R-Fig. 5: Wake (solid lines) and inflow (dashed lines) spectra at downstream 4D (left) and 8D (right) from the single-turbine simulation.

Previous studies have shown that inflow velocity fluctuations at St < 0.3 directly drive wake meandering [5,6]. AGWs enhance inflow turbulent kinetic energy at St < 0.3, as shown by the dotted lines, and thus amplify wake meandering. This result is consistent with the work of Wise et al. [7], who reported that AGWs can increase turbulence levels and strengthen wake meandering.

The second and third highest St in the AGW wake spectra are approximately 0.44 and 0.66. The origin of these apparent harmonics is not unclear and will remain as a topic of future work.

**Revison**

We have added 8D wake spectra from single-turbine simulation in Fig. 8 (as also shown in R-Fig. 5) and discussed the downstream evolution of wake spectra in lines 194-195.

'For both the AGW and non-AGW cases, such a frequency peak becomes less prominent in 8D downstream, because wake recovery has largely weakened shear instabilities at this region.'

(8) Wake velocity recovery. As noted on line 199 the distance to the maximum velocity deficit differs between the AGW and non-AGW cases. It is not however shown that the wake form is Gaussian at this point. The points on 'faster recovery of the mean wake of the AGW' should relate to the distances after this near-wake region. Over that range the AGW wake recovery is faster but it would be useful to bring out this rate of recovery of velocity more clearly. Differences of wake recovery are attributed to two mechanisms: i) stronger wake meandering due to larger-scale turbulent structures, and ii) higher value of TKE in the AGW case. The earlier sections should be modified to support these statements quantitatively including clarification of: the scales of turbulent structures in the onset flows, that the wake meandering is at the AGW periods, the value of TKE of the non-AGW onset flow.

**Reply**

We agree with the reviewer's suggestion and have replotted Fig. 10 (also shown as the left panel of R-Fig. 6) to illustrate the rate of wake recovery in the far-wake region. The far wake is defined as the region where the spanwise wake profile becomes Gaussian. Based on the mean velocity contours in Fig. 9, we identify x/D > 3 as the far-wake region. In this region, the AGW case exhibits a higher recovery rate than the non-AGW case.

R-Fig. 6: Mean streamwise wake deficit along the turbine centerline (left). Wake TKE contours (top-right). Inflow spectra (bottom-right).

As discussed in the original draft, the faster wake recovery in the AGW case can be attributed to two factors: (i) enhanced wake meandering and (ii) higher turbulent kinetic energy. The first factor is supported by the meandering amplitudes shown in Fig. 5 and the inflow spectra in Fig. 8 (and in the bottom-right panel of R-Fig. 6). The increased inflow turbulent kinetic energy at St

R-Fig. 7: Time series (left) and spectra (right) of turbine power for the three-turbine simulations with 4D (top) and 8D (bottom) spacings during the AGW event.

Regarding the reviewer's concern about the operating condition of T3, the time series of exact power (left panel of R-Fig. 7) confirms that T3 remains within its operating range for both 4D and 8D spacings. In our simulations, we set a constant rotational speed of 9 RPM. At low wind speeds, the aerodynamic torque is not enough to overcome generator and drivetrain losses. As a result, the reported power output can be negative, meaning the turbine is consuming electrical power to keep the generator running. We have modified Fig. 12 to the top panel in R-Fig. 7.

We agree with the reviewer that more realistic turbine layouts and more combinations of turbine spacing vs. AGW wavelength should be considered to fully understand the interactions between AGWs and wind farms. These topics will be pursued in future work.

**Revison**

We have added the power outputs for the three-turbine simulation with an 8D spacing in Fig. 11 (as also shown in R-Fig. 5) and discussed the power attenuation in lines 230-235.

Figure 11 shows time series (left) and spectra (right) of turbine power for the three-turbine simulations with 4D (top) and 8D (bottom) spacings. For 4D spacing, the presence of AGWs induces large-scale power oscillations at the first turbine (T1), which are strongly attenuated at the downstream turbines (T2 and T3). For 8D spacing, the attenuation of power oscillations is weaker, and T2 still exhibits visible peaks with a time delay relative to T1. The difference in power attenuation between 4D and 8D spacing is also evident in the corresponding spectra. This behavior is because, as we showed in Fig. 8, the shear instability

mechanism that damps low-frequency velocity fluctuations becomes weaker at further downstream.'

**Reference**

- [1] Allaerts, D., Quon, E., and Churchfield, M.: Using observational mean-flow data to drive large-eddy simulations of a diurnal cycle at the SWiFT site, Wind Energy, 26, 469 492, 2023.
- [2] Churchfield, M. J., Lee, S., Michalakes, J., and Moriarty, P. J.: A numerical study of the effects of atmospheric and wake turbulence on wind turbine dynamics, Journal of turbulence, p. N14, 2012.
- [3] Wu, Y.-T. and Porté-Agel, F., 2011. Large-eddy simulation of wind-turbine wakes: evaluation of turbine parametrisations, Boundary-layer meteorology,
- 138, pp.345 366.
- [4] Stevens, R.J., Martínez-Tossas, L.A. and Meneveau, C., 2018. Comparison of wind farm large eddy simulations using actuator disk and actuator line models with wind tunnel experiments. Renewable energy, 116, pp.470-478.
- [5] Larsen, G. C., Madsen, H. A., Thomsen, K., and Larsen, T. J., 2008. Wake meandering: a pragmatic approach, Wind Energy: An International Journal for Progress and Applications in Wind Power Conversion Technology, 11, pp.377–395.
- [6] Feng, D., Li, L. K., Gupta, V., and Wan, M., 2022. Componentwise influence of upstream turbulence on the far-wake dynamics of wind turbines, Renewable Energy, 200, pp.1081–1091.
- [7] Wise, A.S., Arthur, R.S., Abraham, A., Wharton, S., Krishnamurthy, R., Newsom, R., Hirth, B., Schroeder, J., Moriarty, P. and Chow, F.K., 2025. Large-eddy simulation of an atmospheric bore and associated gravity wave effects on wind farm performance in the southern Great Plains, Wind Energy Science, 10(6), pp.1007-1032.